# The COVID-19 Pandemic in Spain: Experiences of Midwives on the Healthcare Frontline

**DOI:** 10.3390/ijerph18126516

**Published:** 2021-06-17

**Authors:** Josefina Goberna-Tricas, Ainoa Biurrun-Garrido, Carme Perelló-Iñiguez, Pía Rodríguez-Garrido

**Affiliations:** 1Department of Public Health, Mental Health and Perinatal Nursing, Universitat de Barcelona, 08007 Barcelona, Spain; 2Campus Docent, Sant Joan de Déu, Fundació Privada, School of Nursing, Universitat de Barcelona, 08034 Barcelona, Spain; 3Sexual and Reproductive Health Clinic, Parc Salut Mar, 08003 Barcelona, Spain; 4PhD Program in Citizenship and Human Rights, Universitat de Barcelona, 08007 Barcelona, Spain; cperelin7@alumnes.ub.edu; 5Nursing and Health PhD Program, Universitat de Barcelona, 08007 Barcelona, Spain; prodriga16@alumnes.ub.edu; 6Area of Sexuality and Diversity of the Department of Equity and Gender, University of O’Higgins, Rancagua 2820000, Chile

**Keywords:** SARS-CoV2, COVID-19, midwives, health care, health professions, pandemic, qualitative research

## Abstract

Background: Midwives look after women during pregnancy, childbirth and puerperium. In Spain, the first wave of COVID was particularly virulent. There are few studies about the experiences of midwives providing care during the COVID pandemic and very few have been undertaken in the countries of southern Europe such as Spain. This article sets out to take a more in-depth look at the experiences of midwives who were on the frontline of care during the early months of the COVID-19 pandemic as well as to identify new needs and resilience strategies that can help midwives. Methods: A qualitative methodology of phenomenological tradition was used, interviewing 10 midwives from primary care, hospital and independent care. Results: After content analysis, three central categories emerged: (a) cascade of emotions; (b) professional occupation and concern for the women; (c) resisting the day-to-day; resilience and resistance strategies. Conclusions: Despite the difficulties, midwives are concerned about the loss of rights and autonomy and about the increased vulnerability of women. Midwives have become aware of the power they have in their actions both in health management and administration, as well as in the care of women, creating strategies to provide dignified care to their users.

## 1. Introduction

As professionals who look after women during childbirth, whether it be in their homes or in a hospital, midwives play an important role in monitoring and providing care during pregnancy, childbirth, and puerperium. Moreover, and as shown in the bibliography [1,2,3], the professional competences of midwives, their knowledge, skills and attitude can improve the childbirth experience and they can also contribute to the autonomy and empowerment of women during this process, thereby improving results in breastfeeding, the mother–child bond and the women’s mood and level of self-care. Therefore, we can say that midwives play a fundamental role in providing support during the childbirth process.

Various studies have shown that the work of midwives is emotionally demanding in that on a daily basis they find themselves in situations where they have to manage emotions such as anxiety, pain or fear. At the end of 2019, the emergence of the coronavirus SARS-CoV2 led to profound changes in all aspects of healthcare, placing the planet in a state of alarm. On 31 December 2019, the Wuhan Municipal Health [4] Commission in Wuhan City (province of Hubei, China) informed of 27 cases of pneumonia of unknown aetiology, with a common link to a wholesale seafood, fish and live-animal market in the city of Wuhan. According to current evidence, it is calculated that COVID has an incubation period of 5 days (range 2–14 days). The disease initially manifests itself through fever, a dry cough and shortness of breath, with pneumonia being one of the main complications. It was subsequently discovered that the disease can affect the organs and that it is asymptomatic in many cases. The existence of an incubation period during which the disease can be transmitted without showing any symptoms, together with asymptomatic cases, make COVID-19 a highly contagious disease. The disease with clinical manifestations mainly occurs in persons between the ages of 30 and 79, and it is less symptomatic among people under the age of 20. At the start of the pandemic, health professionals were one of the collectives most vulnerable to contracting this disease. Up to the time of writing (14 May 2021), there have been 162,145,693 reported cases of COVID-19 throughout the world and a total of 3,362,875 deaths have been clearly related to the disease. A total of 3,598,452 cases have been documented in Spain [5].

At the start of the pandemic, relatively little attention was paid to delivery rooms since the disease clinically affected the elderly, but it was soon discovered that pregnant women could also become infected, and this created concern about mothers and babies becoming infected by COVID during pregnancy. Some studies have noted that being pregnant increases risk in the event of an infection, but the current vaccination strategy of the Spanish Ministry of Health and Consumer Affairs states that “in general, no greater serious risk of COVID-19 has been detected as a result of being pregnant, beyond the risk conditions that the woman had” prior to pregnancy. Currently available state data suggest that 80% of pregnant women experience the illness [6] in a mild or asymptomatic form, although the possibility of contracting COVID-19 during pregnancy or childbirth has made it necessary to implement new healthcare protocols [7,8,9] in order to provide guidance based on the best evidence available at each moment of the pandemic. The aim of this advice is to guide the practice of midwives and other health professionals who care for pregnant women, women in labor and women who have just given birth, and to advise them on what they should do during the current COVID-19 pandemic.

In Spain, the first wave of COVID was particularly virulent, forcing the health system to urgently adapt pregnancy and childbirth care within a system already collapsed by COVID patients. The high number of COVID patients made it necessary to transfer treatments of other pathologies, which were momentarily put on hold unless they were serious. Non-urgent surgical interventions were suspended, non-essential external consultations were cancelled, but logically, childbirths continued to be attended to. This resulted in delivery rooms being adapted to care for births by women infected with COVID as well as by women who were not affected.

In these circumstances, changes were made to the childbirth care environment and there was an increase in the demand for homebirth care [10]. In Catalonia (one of the seventeen autonomous communities into which the Spanish state is politically organized), there was a recentralization of childbirth care, and some hospitals closed their delivery rooms in order to allocate the maximum number of beds to COVID patients. Obstetricians and midwives were sent to care for patients with COVID because of the fact that they had qualified as doctors or nurses prior to specializing in their particular area. Childbirths were concentrated in a reduced number of hospitals where the delivery rooms had been specially prepared and isolated to a certain extent from the other areas of the hospital where COVID patients were being cared for.

On 4 May 2020, the International Confederation of Midwives published a document seeking to alert the world health authorities to the lack of attention to women’s rights in matters relating to childbirth care that resulted from this health crisis, and to the neglect of midwives and other health professionals who attend to women at childbirth [11].

Today, there are several published studies on the experiences of patients during the COVID-19 pandemic [12], as well as on healthcare workers infected by COVID-19 [13], and even some on women during pregnancy and the postpartum period [14]. Likewise, the bibliography includes research work about the experiences and consequences for the mental health of health staff who worked and who are working during the COVID-19 crisis [15]. However, there are few studies about the experiences of midwives providing care during the COVID pandemic and very few studies have been undertaken within the European context, particularly in the countries of southern Europe such as Spain [16]. Accordingly, this article sets out to take a more in-depth look at the experiences of midwives who were on the frontline of care during the early months of the COVID-19 pandemic. A secondary objective is to try and identify and reflect on new needs and resilience strategies that can help midwives with decision making and assist them in managing the situations and consequences that have derived from the pandemic.

## 2. Materials and Methods

### 2.1. Design

In developing this study, we used a qualitative methodology that took a descriptive approach based on the phenomenological tradition. In this regard, our belief is that a qualitative methodology allows naturalistic interpretations and approaches to the research subject, and according to Denzin and Lincoln [17], this “means that qualitative researchers study things in their natural settings and try to make sense of, or interpret, phenomena in terms of the meanings people bring to them”. In our case, we will be guided by the Husserlian approach which views phenomenology as “a paradigm that seeks to explain the nature of things, the essence and the veracity of phenomena. The objective that it pursues is to understand the experience lived in its complexity […] and the meanings centred around the phenomenon” [18].

Experiences as a unit of analysis provide “a way of interpreting, assessing and making sense of reality, whilst reflecting the unity of socio-cultural and personal aspects” [19]. For this reason, we have aimed to understand and take an in-depth look at the experiences of midwives who have found themselves on the frontline of healthcare because of COVID-19.

### 2.2. Context of the Study

The study was carried out in Spain, and specifically in the autonomous communities of Catalonia and the Balearic Islands. Spain is divided politically into seventeen autonomous communities which have full powers on healthcare matters. Catalonia is situated in the northeast of the Iberian Peninsula, and it has 44 publicly owned hospitals and 27 private ones, which offer childbirth care. Pre-natal monitoring is carried out through the primary healthcare network which has care centers distributed throughout the territory; midwives are the professionals of reference for monitoring low-risk pregnancies [20]. Administratively, Catalonia is divided into four provinces with Barcelona as its capital, and the city where the hospitals of greatest complexity are located. The Balearic Islands is an autonomous community formed by a group of five islands in the Mediterranean Sea, the capital of which is the city of Palma de Mallorca. There are 7 referral hospitals in the Balearic Islands and a primary care network covering the 5 islands, which monitors pregnancies [21]. Even though both communities were affected by COVID, Catalonia was one of the most affected because of the high population density around the city of Barcelona, while conversely, the insular community of the Balearic Islands was one of the least affected in Spain during the initial surge of the pandemic. At the end of April, the Cumulative Incidence per 100,000 population was 158.29 in Catalonia and only 23.75 in the Balearic Islands [22].

### 2.3. Selection and Characteristics of Participants

Interviews were held with active midwives. Some theoretical selection criteria were established, and these were used to cover the largest possible number of profiles of midwives. The common inclusion criterion was that they had to have been on the healthcare frontline during the first months of the pandemic, and we also sought variability in terms of the work center (hospital, primary care or homebirth care), age variability (29 to 50 years old), and we also looked to include midwives who had taken ill with COVID-19 as a result of becoming infected through their professional activity. We also included midwives who were pregnant at the start of the pandemic so that they could express their concerns as future mothers in addition to their professional viewpoint. Accordingly, interviews were held with midwives with the following profiles:Hospital midwives who continued to provide care in delivery rooms during these months;Midwives who provide care at home births;Primary care midwives who continued to provide care throughout these months;Hospital midwives at centers where the delivery rooms were closed and who were transferred to care for COVID-19 patients;Midwives in hospital or primary care sectors who contracted COVID-19 while carrying out their work;Hospital or primary care midwives who were pregnant or who became pregnant at the start of the pandemic.

The participants were identified using two mechanisms: firstly, known midwives were contacted by telephone using the personal contacts of the researchers, and “snowball sampling” was used to identify midwives who might meet the previously established theoretical criteria. The second strategy used was an advertisement on the website of the Research Group inviting midwives who would like to be interviewed. A total of 10 midwives were interviewed and their profiles are shown in Table 1 below (the names have been changed to preserve the anonymity and confidentiality of the participants).

### 2.4. Technique Used to Gather the Information

An individual semi-structured interview was the technique used to gather information. This type of methodological technique is very useful since it is understood as “a communication process that occurs in previously negotiated and planned meetings between subjects” [23] in order to take a closer look at the experiences surrounding the studied phenomenon. To prepare the interview outline (Figure 1), we started with a bibliographical search of the literature published up to the time that we started our research on the experiences of professionals in the care of childbearing women with COVID-19, and also in relation to vulnerability within the area of obstetric care [24].

Due to the pandemic situation, the interviews were held online between June 2020 and March 2021 using the Collaborate application hosted on the University of Barcelona intranet (thereby guaranteeing the confidentiality of the data gathered). All the midwives interviewed were offered the possibility of participating with their camera switched on or off during the recording in order to preserve their image rights. The duration of the interviews ranged from 22 to 60 min.

### 2.5. Ethical Aspects

The research was approved by the University of Barcelona Bioethics Committee (IRB00003099). The objective and ethical considerations of the study were explained to all the participants by e-mail, and they were also sent the information and an informed consent form by email, which they signed and returned to the Main Researcher before each of the interviews in the case of the participants where the contact was by telephone. In the case of participants who were contacted through the website, the information was included on the website and the participants gave their consent to be interviewed on the website itself, when they provided their personal data and their contact email.

### 2.6. Criteria for Methodological Rigor

We considered the list of questions contained in the Standards for Reporting Qualitative Research (SRQR) [25] throughout the development of the study and in the drafting of the final report. Furthermore, we used the following quality criteria in accordance with Calderón [26]: (a) “epistemological adequacy”, that is to say, reviewing the formulation of the research question and the coherence of the process; (b) “relevance”, since the need to know the experiences of frontline midwives during the pandemic was clearly justified given that their knowledge will allow us, on the one hand, to take an in-depth look at the journey that they have been on, while on the other, they will generate situated knowledge that will be disseminated in the scientific field and used to make proposals for improving care. Similarly, the (c) “validity” criterion is not intended to be understood in terms of statistical probability but rather in terms of relevance and interpretivism, and so, an appropriate process has been sought for the selection of participants and to guarantee rigor in the analysis in order to know meanings and look for in-depth generalizable explanations from a logical point of view that are transferable according to the contextual circumstances in which the research was carried out. Finally, (d) “reflexivity”, that is, it is also important to recognize the position of the researchers both as midwives and researchers who are also immersed in the scenario of the COVID-19 pandemic.

### 2.7. Data Analysis

The interviews were recorded in MP4 format using the application Collaborate, which is hosted on the intranet of the University of Barcelona, which was also used to conduct the interviews. The interviews were subsequently transcribed by the Main Researcher and a collaborator who was external to the research. They were then analyzed using qualitative research methods based on the criteria of Taylor and Bogdan [27]. The first step consisted of a careful reading of the transcriptions to obtain ideas and intuitions, and the second step involved categorizing the data into information units and grouping them into categories based on similarity, which responded to the objectives of the study, while the data were relativized during a third phase to contextualize them. The codes and categories that emerged were discussed by all the members of the research team.

## 3. Results

After analyzing the transcriptions, various aspects emerged in relation to the experiences and expectations of the midwives who were interviewed. Some aspects were based on subjectivity: the emotions they felt. An event as disruptive as the COVID-19 pandemic has several effects at a mental and emotional level. During the interviews, the midwives told the researchers about their emotions and experiences during the first months of the pandemic. In addition to personal and family aspects, the midwives had to carry out their work on the front line, and accordingly they had a responsibility to look after and accompany women who were pregnant or in childbirth. In Spain, midwives must first qualify as nurses before specializing, and this professional nursing qualification meant that some hospitals transferred them to look after COVID patients in general hospital wards. The midwives felt a conflict between their personal feelings and their sense of professional duty. The case of midwives who were pregnant at the start of the pandemic is also worth mentioning here, as they had to deal with the risk of infection, the doubts they had as to whether they should request leave from work and the frustration that stemmed from the feelings about pregnancy and childbirth that they had imagined before the pandemic. Given the important difficulties of the moment, the majority of the interviewed midwives developed resistance and resilience strategies, and these can be observed in the analysis.

We have created the following categories based on our analysis of the transcriptions (Figure 2): (1) cascade of emotions, (2) professional occupation and concern for the women, (3) resisting the day-to-day; resilience and resistance strategies.

### 3.1. Cascade of Emotions

The midwives mentioned several emotions throughout the interviews: fear; uncertainty and insecurity; concern, distress, sadness and anxiety; confusion; loneliness and despair, anger and impotence. Midwives experience these emotions when exercising their profession, both at the level of their direct experience as professionals and from a humanity perspective, and also because of the relationship that they establish with the patients that they look after:

#### 3.1.1. Fear

Fear was a recurring emotion among all the interviewed midwives. This fear affected them both with regard to themselves—the very fear of getting infected as well as fear of the unknown—and as citizens of a country in a pandemic situation. The fear of infection was very present during the first months of the pandemic.

“Well, at the beginning I was a bit scared, like everyone I guess, I sort of freaked out when the state of alarm was declared nationwide, it was a bit shocking. I’d never been in a situation like that in my whole life and it was a bit shocking especially because of the fear of becoming infected”.(Aurora)

The fear was expressed in their everyday actions, just like all other people, but being a frontline health professional meant that this fear was expressed with a greater intensity in their day-to-day work. As we have said, during the first wave of the pandemic in Spain, births were centralized in certain hospitals and many delivery rooms were closed. Midwives in those hospitals where delivery rooms were closed were obliged to transfer to wards for COVID patients where they worked as nurses. This situation created a sensation of fear linked to the disease and a fear of infection, but most particularly a fear of a change of professional role. As midwives, their knowledge in treating infectious diseases was limited or out of date:

“Of course, we were scared at first, because we hadn’t worked as nurses for a long time...”.(Alice)

These emotional expressions shaped the professional paths of the midwives in the sense that they had to readapt their everyday practices, and this resulted in great uncertainty and insecurity about that process.

#### 3.1.2. Uncertainty and Insecurity

The midwives also spoke about the uncertainty that they experienced, especially at the start of the pandemic. The information that they received at the healthcare centers where they worked and also through the media was confusing. They were overcome by a sensation of unreality, which increased the uncertainty.

“Everyone was a bit lost at the time, didn’t know what to think, even when we knew what was going on in China or Italy, in my opinion nobody could have imagined... It felt like we were immersed in a virtual reality”.(Nancy)

The uncertainty was also linked to a lack of foresight, and this was especially clear in the lack of personal protective equipment (gowns, masks, protective screens etc.), which reached some hospitals after a significant delay.

“[...] because it’s a whole new situation and you’re aware that there won’t be any equipment available...”.(Alice)

The pandemic hit all the autonomous communities of Spain very hard, but it particularly impacted the heavily populated metropolitan areas. Many Service Heads and Hospital Supervisors took ill, and this resulted in the midwives finding themselves on board a boat without a captain or any fixed direction. In some cases, the lack of direction increased this sensation which was already uncertain in itself.

“[...] then for a while there was nobody in charge, leading the whole process, it seemed that we were being isolated, and then the thing with the PPE (Personal Protection Equipment), it was not clear what we were supposed to do or when and how we had to wear it [...]”.(Elisa)

“[...] we had the feeling that we were drifting because nobody was directly responsible for coordinating our response”.(Jennifer)

The transfer of some personal protection equipment to services in COVID wards generated feelings of insecurity in them. Professionals who had spent years away from general nursing in a hospital ward suddenly saw how they had to return to this role that they had forgotten, or which they had never actually performed.

“Other nurses were transferred to another floor. Obviously, some of us had not worked in a hospital floor unit for many years, taking care of the patient’s hygiene with the nursing assistants, with... You know, usually we don’t take care of the patient’s hygiene in the delivery room. We clean the mother, but they are self-sufficient... And then, of course, all of a sudden, you have much older patients. And obviously you gotta deal with their nappies, well... it’s a huge change of role, the way you take care of the patient, it’s a totally different patient”.(Aurora)

The lack of protective equipment also caused this feeling of insecurity. The professionals were totally exposed to the virus because of a lack or shortage of equipment necessary for their individual protection. This situation was generalized during the first few days, but personal protective equipment started to arrive gradually, and the feeling of insecurity began to dissipate.

“[...] it’s true that as things were getting better, we had more equipment, more PPE, we were calmer, right? You feel safer doing your job”.(Elisa)

In the case of midwives who were pregnant, this insecurity manifested itself in the continuous search for information and their realization that there was a shortage of evidence about the consequences that coronavirus could pose during pregnancy. This insecurity was translated into fear and insecurity as in the case of this midwife who already had risk factors associated with her pregnancy before the pandemic:

“Besides this, I looked up some things on the Internet and I also searched for articles about the pandemic in China but there wasn’t too much information about the consequences that infection could cause during pregnancy. It was initially thought that infection did not pose a threat for pregnant women but with the passing of time, we started to see that some studies associated it with a higher risk of preeclampsia and a greater risk of other pathologies, and since my pregnancy had been diagnosed as high risk, well I got even more scared”.(Aurora)

These circumstances were essential factors with regard to facing up to a reality that was changing, and which also became more complex as the days went by. This feeling of insecurity created even greater worries and concerns about the actions and personal and professional activities of the midwives who were interviewed.

#### 3.1.3. Worries and Concerns

Midwives voiced worries and concerns about a large number of issues. Their concern was compounded by aspects such as the fact that the new data were only being published in scientific studies and articles “in dribs and drabs” because of the lack of information, certainty and proof. This left the midwives without any information or knowledge that they could use as a means of support.

“Maybe it’s true that my healthcare practices are a bit unorthodox in this sense. I don’t completely trust studies that undermine what’s going on and underplay the risks, maybe I feel a bit insecure in this regard”.(Nancy)

The lack of any foresight, the absence of a horizon, and the lack of any certainty about the future also contributed to this situation. Life was limited to “surviving”, to simply taking one day at a time:

“Because of the uncertainty, because I don’t know what the situation will be like in a year, I find myself struggling to cope with my family life and keep working at the usual rhythm, because I like to give my best at work, and having to slow everything down takes its toll, because I’m a very active person, but I think I’m managing, trying to keep calm. Now I’m just in survival mode”.(Bárbara)

These insights allow us to see how professional decisions had to fit in with personal decisions, and very often this created a paradox for health professionals who have to look after others while simultaneously having to look after themselves in settings characterized by worry and concern.

#### 3.1.4. Distress, Sadness and Anxiety

In their interviews, the midwives referred to the anguish that this situation caused them. The uncertainty was converted into distress and a lack of safety:

“[...] you’re working there coping with the anxiety of not knowing if you’re properly equipped, of not knowing if you’re dealing with infected people... perhaps you’ll take the virus home with you, and then you realize that the people in charge are also sick at home, the situation was a bit...”.(Jennifer)

This distress was greater among those midwives who were redeployed to COVID wards. The situation of fear that we mentioned earlier regarding the new professional roles that they had to take on also highlighted their uncertainty about how to perform these new roles properly, and this caused distress and anxiety.

“And then the anxiety of having to administer an amount of antibiotics you’re not used to dealing with, of using medicines that you have not known even how to dilute for a long time and giving them to elderly patients who are not only suffering from COVID but who also have their base pathologies, many medicines you don’t know how to manage”.(Aurora)

The actual vulnerability that they felt when faced by the pathology, the social situation, the work situation, etc. created a feeling of sadness in the professionals and in some cases, they even feared that they might end up suffering from depression.

“I felt weaker with a constant need to cry, I was helpless with a feeling that the situation we were going through was chaotic because we didn’t know the extent to which our unconscious and memory would be affected, the after-effects, the “post-traumatic stress” so to speak”.(Jennifer)

These experiences offer an insight into the level of suffering and psychosocial unease that the interviewed midwives had to face when working during the most complex stages of the pandemic, which had major consequences for their mental and emotional health.

#### 3.1.5. Confusion

Midwives who took ill with COVID and who had to be admitted to hospital as patients found themselves experiencing confusion, especially at the start of the pandemic. The change from a professional to a patient stricken with an unknown disease and seeing the colleagues who were caring for them with the same fear that they had experienced, plus the lack of operating guidelines, simply added confusion to all the other emotions that they felt. The sensation of unreality caused by the uncertainty generated confusion about all the measures taken in the health area.

“It’s an illness where doctors don’t even see you, where they check on you from the door, they show fear when using their stethoscopes on you, everyone is really scared. I think the first few weeks, when everyone was a bit lost, were mostly defined by a sensation of pervading fear, of confusion, as experienced by the workers (we are not really in touch with the doctors) but especially by the nurses, supporting staff such as the cleaning personnel and the nurses... a lot of confusion, fear, they were totally baffled by a situation they had never encountered before”.(Nancy)

Those who continued to work and who did not become infected were often puzzled by social reactions, as in the following case of a midwife who explains how she felt when people would come out to applaud at 8 o’clock every evening:

“It was totally like being in a movie. Society, the pervading fear, gathering at the balcony every day at eight o’clock to clap for the health workers, everything was a bit... weird”.(Aurora)

The social recognition and public support shown by people when they came out to clap each evening was perceived as a positive gesture although it was also somewhat confusing as the previous midwife mentioned. A certain discordance was perceived between the act of applauding the daily efforts of health workers and the dystopian scenario in which this occurred at a time when everybody was in lockdown because of the pandemic.

#### 3.1.6. Distrust Because of the Uncertainty of Decisions

Midwives started to distrust some of the decisions adopted by health centers because of contradictory information, incoherencies, rushed or emergency decisions before an unknown situation. An example of this is the case of one midwife who contracted COVID and who had to be admitted to hospital:

“They tried treatments that were, well, a bit experimental, to see how people responded to them. You’re scared because doctors you’ve always trusted to cure you, well, you feel like a guinea pig, that they were trying things on you to see how you respond to the treatment”.(Nancy)

Moreover, a certain confusion arose about the contradictory aspects of the social reality related to the political decisions that imposed lockdown and restricted mobility:

“I don’t feel calmed yet, and if my parents do manage to come to Catalonia, which I’m not sure they’ll be able to, because on the one hand the authorities are telling us that we have to keep two meters apart from each other, right? But on the other hand, the planes are crammed with people, they don’t even leave an empty seat between passengers. Well, it seems that certain things are allowed and others... I don’t understand what all this means, but I will only allow my parents to come to see me, I won’t be meeting any friends for a long time, a year maybe, I don’t know, when the situation is a much more under control”.(Bárbara)

There was also distrust among the various professionals working at the hospital, such as gynecologists and midwives. At the start of the pandemic, many health centers took the decision not to allow companions and partners to be present at births. This created some conflict and led to complaints by women who were about to give birth; the decision was later reversed following complaints and claims by the patients. The collective of midwives positioned itself, for the most part, on the side of the women, and this generated a certain level of unrest and even some misunderstandings:

“And this, on an internal basis, has brought us a lot of trouble and misunderstandings, because many of our colleagues in gynecology were convinced that it was us, the midwives, who were behind all this, and the truth is that it had nothing to do with us, the women organized themselves and created a network to make it loud and clear that “we don’t want to be part of this””.(Antonia)

The feeling of doubt and uncertainty occurred at all healthcare levels. Seeing the heads of service unable to make any clear decisions multiplied the doubts at all levels:

“It seemed impossible, even some heads of service had their doubts... we were not sure if it was necessary to cancel so many appointments. And they told us that it was, “yes, everything must be cancelled””.(Nancy)

Doubts also arose because of the insufficiency of the knowledge about the disease itself and the lack of knowledge about the ways and means of infection during the early days of the pandemic. The lack of knowledge about asymptomatic cases facilitated situations where the midwives themselves could infect their family members without knowing it:

“[...] Seeing that we were getting sick, not sure how we became infected, not sure if we were properly equipped, because otherwise nothing made sense, you felt fine, but the folks at home were getting sick”.(Jennifer)

Faced with this constant perplexity about the way of deciding the context medically and politically, the feelings and emotions of the midwives intensified, and many professionals opted to stay on their own or to isolate in order to cope with the pandemic whilst protecting their loved ones from getting infected.

#### 3.1.7. Loneliness

The professionals felt alone both professionally and personally. Faced with the dangers they were exposed to, the professionals had to take measures that meant isolating themselves from their family members and their friends. This type of precaution increased their sensation of loneliness. Some of them sought tools to help mitigate this feeling. This is the experience of one midwife who was pregnant at the start of the pandemic and who was granted sick leave because of the risk of contagion; her partner was a frontline doctor:

“[...] I lived with my husband and he had to go and stay in a hotel room, he had to stay there for a month and a half. If we add to that the fact that I couldn’t see my gynecologist, because all the visits were performed over the phone, that my family was also away ... It required fortitude, personal strength, writing a diary where I could share my emotions was a great help to me”.(Barbara)

However, professionals also perceived and suffered the loneliness of their patients.

“At that moment, the situation required us to act as nurses, midwives, to provide the best care to the people who needed us the most, but, of course, I have also spoken with some of the women... We cannot make the same mistakes again and... I think that I must do something for our women patients, who also had to go through this alone”.(Alice)

This series of emotions and experiences show a need to look at care in a collective manner, and this involves looking after the professional team at the same time. Midwives, like other health professionals, are carers, but these extreme situations have shown that they also need to be cared for themselves. When identifying the importance of collectivizing care, it is important to consider the fact that women are the main agents of such care, and this situation is particularly evident among midwives. This resulted in the midwives who were interviewed feeling that they were putting in a double working day, providing care in their professional role in addition to their role as carers at home.

#### 3.1.8. Despair, Anger and Impotence

The absence of any horizon and the sense of uncertainty about the future led to a feeling of hopelessness and despair.

“Hopelessness and rage, because of the uncertainty of not knowing exactly what was going on or of not having...”.(Bárbara)

In some cases, the interviewees referred to complicated family situations, as in the case of this midwife who was separated, and who was not able to see one of her children throughout the time of the pandemic:

“[...] COVID has had a great impact on my life, because I work in the health sector, my ex-husband convinced my daughter not to come to my home because I was working at the hospital, and so I spent three months without seeing my daughter”.(Antonia)

In their day-to-day work activities, midwives are used to dealing with the unexpected things that can arise during the process of providing care during pregnancy, childbirth and the postpartum period, but the uncertainty was such because of the interference of the pandemic that the midwives transformed this situation into anger and impotence.

“[…] And I think that’s shame it was not allowed, the personals who are in the front line have to be endowed with the greatest team, it makes me very angry”.(Bárbara)

Finally, in this process of the pandemic, the feelings of impotency and anger also intertwined due to the experiences that the midwives had to endure in their new roles and activities, which not only affected their own lives but the lives of those around them as well.

### 3.2. Professional Occupation and Concern for the Women

In addition to important aspects related to the emotions experienced, there were categories related to the professional development of their activities as midwives, as well as their concern for the emotions and rights of the women under their care:

#### 3.2.1. Invisibility

The invisibility of delivery rooms in the way that hospitals managed the pandemic is a clear symptom of everything that we have outlined thus far. At the beginning, maternity services were not a focus of attention because of a lack of knowledge about asymptomatic cases and the belief that the illness only affected older people:

“[...] but we felt a bit as if we’d been forgotten, you know? Because the crisis committee would meet on a daily basis and these meetings were reported, you know? I had access to those briefs via the trade union’s WhatsApp group, and well, the delivery room was not even mentioned”.(Antonia)

This invisibility and the idea that the illness did not affect pregnant women or delivery rooms resulted in incoherent actions, and this caused midwives, and in particular those in management positions, to reflect later:

“A very uncommon situation, because in all those meetings between the hospital management and the department heads, we didn’t take any precautions, we did not respect the safety distance or wear any masks...”.(Nancy)

This series of actions implemented at the start of the pandemic can be seen as part of a chain of errors, which were identified and corrected after the first weeks, but which affected maternity units and the daily work of the midwives.

#### 3.2.2. Changes to Routines and the Economic Crisis

The changes of habits and routines that they were forced to make were a consequence of the preventive measures adopted to contain the virus and also the result of changes of roles (nurses working in COVID wards). This was compounded by the economic crisis that affected society as a result of the lockdown and the cessation of many economic activities. On occasion, these changes of habits were quite neutral, but in some cases, they reflected a level of precariousness in the work or the families of the midwives or their partners, and this was accompanied by poor perspectives for the future:

“I think that many workers will be made redundant and will have to take more than one job or come up with a plan if their partner is on a temporary layoff, and you don’t have the expected income. Then, I think that we’ll see a rise in work mobility”.(Aurora)

The midwives also perceived changes in their professional activity:

“[...] And considering our job in primary healthcare, I also think that we’ll resort more to virtual tools, more phone calls, more online group learning, things like that will start to be implemented. As for birth assistance, I don’t think that will change much, the emergency services will be pretty much the same too, as I see it. Had I been working during the pandemic instead of being on a maternity leave, I would been relocated, because the hospital where I worked closed and became an ICU unit”.(Aurora)

The professionals also suffered labor exploitation because of new changes to timetables and services. They were required to make changes, relocate and there was an increase in the healthcare burden. On this issue, one midwife told us how public and private care services were reunified because of the pandemic situation:

“What is more, we also had to deal with the centralization of birth care... the private sector services were transferred to public hospitals, and so we had to deal with more patients, midwives had to face an increasing workload”.(Antonia)

This was especially evident in the case of those midwives who were redeployed in wards for COVID patients.

“At our hospital birth-care services were transferred. First, they asked the midwives if they wanted to go, but afterwards, they told us that we had to go, that is, we were not forced to go, but it was clearly a command, you have to go, or they need you there, and...”.(Alice)

In the subjective view of the midwives, this precariousness was also seen as a lack of care for the women because of the move from in-person to online activities. Some midwives felt that this amounted to a poor level of care insofar as the women felt that they were not well accompanied.

“Nurses in general and midwives in particular do their job by the patients’ side, and in the case of pregnant women, midwives stay with them every step of the way, an assistance and accompaniment that in this case must be done via phone call”.(Nancy)

These changes in healthcare habits also resulted in midwives perceiving a lowering of the quality of support health services during pregnancy, childbirth and the postpartum period.

“Telephone visits or video calls, cancelled visits, no childbirth preparation classes, no pap smears, nothing. They only saw pregnant women. They have continued to act as midwives, but the management of resources has changed due to the need to schedule home visits instead having pregnant women come to the hospital... PCRs have also been performed on their birth partners”.(Sara)

All this realigning of functions and roles as part of the new reality caused by the pandemic generated the feeling that there was a drop in the quality of professional care and a loosening of the bonds with the women who were being cared for in the usual maternity centers. This led to the feeling that the professional actions of the midwives were inefficient.

#### 3.2.3. Need to Improve the Work/Life Balance

Considering the bad experiences that they have had, such as the never-ending shifts with no substitutions, etc., they see a need to establish work–life policies that will help to balance their personal and professional lives. These changes should occur in all social aspects for both the midwives and the women patients.

“That’s what I think sometimes that our society has its shortcomings, especially when it comes to creating support networks and resources, so that they help us balance our work and family life, until the kids are three years old”.(Aurora)

This is particularly evident in the case of those midwives who gave birth during the early months of the pandemic, and who experienced major difficulties in trying to return to work with small children as they did not know whether they could leave them at the kindergartens due to the pandemic situation, and neither could they leave them with family members (grandparents, etc.) because of the risk of contagion among older people or difficulties to travel:

“My husband and I are alone here... My mother-in-law wants to come from Colombia to live here and help me with the child, but I don’t know if they will let her come... My parents don’t live here either, I’m not sure it’s safe to leave the baby at the nursery... I will have to find someone to take care of the child at home, but these days it’s not easy”.(Bárbara)

“This is my greatest fear now, I’m worried about who will take of the baby, but there aren’t too many options, because you think “Well, I could leave him at the nursery”, but they don’t know what to tell you either. Of course, it’s not so easy to find a reliable person you can trust to take care of the baby. And leaving him at the nursery is also a risk, because the children also get sick there. If the baby has a fever, they’ll call me and say to me: “Hey, come pick him up”. And what can I do then? And this not just my problem, but... This is what I’m most concerned about now and what keeps me awake at night”.(Aurora)

The tasks associated with caring for the family were among the chief problems to be identified following the introduction of the mobility restrictions and the closing of the spaces that traditionally provide support for families and help in raising children. This situation also highlights the need for greater social and community support within the context of childcare and parenting, especially in the case of professional women, such as midwives, working on the healthcare frontline.

#### 3.2.4. Concern for the Feelings of Expectant Mothers

The fear that the midwives felt for themselves was increased, if that is possible, by the fear they saw in the women that they were caring for. This situation was one aspect that midwives found very difficult to cope with, and they stressed the fear that pregnant women felt when using healthcare facilities, which had gone from being seen as safe places for childbirth care to being considered unsafe for pre-natal and childbirth processes because of the possibility of infection. On occasions, the home became the best place to accompany pre-natal women and assist at childbirth, although the idea of midwives entering homes was also a focus of concern insofar as they might bring the infection with them.

This was even the case for those midwives who went to the homes of expectant mothers to attend to home births:

“Pregnant women were afraid of us when we visited them at home”.(Maria)

In the hospitals, the midwives saw how some expectant mothers refused to visit emergency obstetric clinics during the first days of lockdown if they had any problem for fear of becoming infected in the hospital:

“And then the virus spread, and of course some people who showed symptoms got worse, their condition got more serious and they had to be admitted to the intensive care unit and eventually died, there were people who were admitted to the ICU, there were women who only went to the hospital when their condition was really serious, but others came and were terrified, well, they didn’t know if they were supposed to come... and, well, of course, we’re an emergency service, a very specific emergency service that does not exclude the possibility of our patients being infected with the virus. Many patients were admitted to the hospital in a critical condition as a result of COVID. Everything was a bit chaotic, especially during the first weeks”.(Antonia)

#### 3.2.5. Violation of Rights

The pandemic saw a regression within the area of sexual and reproductive rights, where the medical authority prioritized the safety of protocols, but the scarcity of scientific evidence saw obstacles being placed in the path of respecting women’s rights and autonomy. During the pandemic, there was a deterioration in relation to decision making and the empowerment of women. This created enormous concern among the midwives as the struggle to gain respect for these rights has been a long and tough one, and the midwives feel that many rights have been lost very quickly because of the pandemic. Autonomy and decision making based on informed consent have given way once again to patronizing attitudes.

“Yes, that was already happening, and all these studies that were being published of how it could affect pregnant women in the 40th week of gestation. But now... the way you tell these women that labor must be induced, the information given to them by their gynecologists, and how their choice is limited, informed decision making... So yes, I believe that the right to choose has not prevailed somehow in these circumstances”.(Elisa)

On the other hand, protocols that ignore ethical and social aspects have been introduced.

“In the past you did what the doctor told you to do without even blinking, regardless of the patient’s opinion; now for a time we’ve been working hard to develop a healthcare paradigm where it is the patient who makes her own decisions and who can choose, has the right to know what’s going on with her health at all times, can ask to see her clinical record and can make her own decisions. For what I’ve seen, when facing uncertain situations, we tend to go a bit backwards and bring back the old biomedical and patronizing healthcare practices”.(Nancy)

#### 3.2.6. Safety versus Autonomy—How to Approach the Dilemma?

The dichotomy between safety and autonomy and freedom in decision-making has taken its toll on care for mothers. Safety (in relation to the disease and the virus) has been put before fundamental rights such as in the case of patients’ autonomy and decision making. This has been a source of worry for professionals as it could lead to a slippery slope.

“I’m in favor of letting women give birth how and where they feel more secure. If a woman gives birth at home, provided that hers is a low-risk pregnancy and that she had made the necessary arrangements with qualified professionals, and also that both she and her partner have tested negative for COVID, I don’t see why there should be any problem with her giving birth at home... but now...”.(Bárbara)

Safety was a constant concern and working directly with COVID-19 patients together with an increase in the demand for homebirths created a tense situation for the midwives, especially those who were working in both hospitals and in homecare settings. This made it necessary to take important decisions regarding their work and to assess the safety of the care, on the one hand, and respect for the autonomy or the women on the other. In some cases, the midwives took drastic decisions, such as ceasing to attend to home births because of the dilemma between safety and freedom of choice:

“We’re a group of four midwives that help to deliver babies at home in low-risk pregnancies; during the pandemic we have not been able to do our job because, well, we also work at the hospital and we thought that it might add just another risk factor... We decided we would not take any chances. But we’ve seen an increasing demand for home births, and we’ve transferred these requests to other colleagues who were willing to do it. Many of them have been assisting far more births than usual, and even had to decline in some cases due to the rising demand, but... for us it’s been like this, working at the hospital, it didn’t seem convenient to have us assisting births at home”.(Antonia)

#### 3.2.7. Vulnerability and Obstetric Violence

When carrying out their duties, the professionals were worried to see how the women who were more vulnerable before the pandemic became even more vulnerable as a result of the protocols established at the health centers.

“All visits were cancelled even for women with high-risk pregnancies”.(Sara)

The professionals themselves also saw an increase in their own level of vulnerability as a consequence of their exposure to the risks of the disease and the lack of protection. In other words, the very persons who should have been providing safety became vulnerable agents themselves.

“We had to work with plastic bags on. We didn’t have protection equipment, even coats were scarce”.(Jane)

“They didn’t give us the first coats and FP2 masks until we were three weeks into the pandemic, that is, until then we didn’t wear any masks as we were supposed to, I was scared”.(Antonia)

There was also some mention of obstetric abuse in the decisions of medical staff that worried and confused the midwives.

“I believe that many births have been delayed and many unnecessary inductions carried out, as many pregnant women have not delivered their babies spontaneously, their births were induced. It was a very stressful situation”.(Antonia)

Vulnerability was a central factor that the interviewed midwives identified during this social–healthcare crisis caused by COVID-19. The lack of concern for the humanization of care because of the urgency of the early weeks in relation to care for pregnant women, the precariousness of the work conditions and of the equipment to protect health professionals against the virus resulted in practices that can be classified as obstetric violence. These included aspects such as inducements without any medical justification. All this dented the morale of the women and the midwives.

#### 3.2.8. Dehumanization

The COVID-19 pandemic brought a series of consequences of an emotional nature, and one of these, which tallies with what the interviewed midwives had to say, was the dehumanization of the life cycle. Birth, death, illness and the relationships between health professionals, were all greatly altered and dehumanized during the pandemic. The tools and materials to avoid contagion, the care protocols and isolations, were some of the most explicit manifestations of the dehumanization of both patients and midwives.

The protocols established by some hospitals were bereft of any humanity (people dying alone and giving birth alone). The midwives suffered emotional problems because of this dehumanization of their work. This was especially intense in the case of midwives who had worked as nurses in COVID wards and who had to deal with cases of people dying alone.

“Well, psychologically speaking, I don’t think I can do it, I suffered from anxiety a few years ago, I’ve been better since, but it’s a lot of pressure to take on, I don’t know, maybe I’ll give it a try, but I don’t think so, it’s taken too much of a toll on me to see people all alone, and there was nothing you could do, and they died alone, and, psychologically, I cannot bear it”.(Alice)

This dehumanization was also present in delivery rooms where the process of providing care during pregnancy, childbirth and the postpartum period was denaturalized and turned into a purely biological procedure. In some cases, the midwives did not provide the response that was expected of them because of their emotional problems, and they suffered upon realizing this.

“Well, maybe, precisely due to that fear, I didn’t show the empathy the situation required”.(Elisa)

The need for protection in the name of safety increased the perception of a dehumanization of childbirth care. Midwives have felt a lack of closeness:

“Screens and protective equipment (with all the effort it had taken just to allow the midwife to be by the women’s side throughout the process...)”.(Jane)

### 3.3. Resilience and Resistance Strategies

Faced with this stressful and disruptive situation, which triggered several ethical conflicts, calling the values of humanized care into question, midwives developed resistance and resilience strategies.

#### 3.3.1. Companionship and Mutual Trust

Teambuilding and companionship have been fundamental both in terms of feeling supported when faced with so many negative emotions, and in order to be able to carry out their professional tasks with the maximum level of quality in spite of the circumstances.

“Yes, and it was an opportunity to develop teamwork. We had a problem and we had to create circuits; there were three of us on our first watch and we had some time to spare, so we decided to go to the operating theatre and ask: “How are you doing it? How are you dressing up? How do you take your PPE off?”. We created our own circuit and with everybody’s help we generated this whole network of protocols and procedures that were required to do our job optimally”.(Jennifer)

Although they expressed distrust in relation to the political and health authorities, they did place their trust in the actions of the professionals, and this strengthened the bonds between the members of the team.

“Personally, I faced the whole situation with a complete trust in the health system, in its professionals, in our capacity to make it through...”.(Nancy)

“As I’d been working closely with my colleagues from the emergency services for many years, we’ve always had a close relationship, and well, now, with all this COVID thing, it got even closer... Because I had so many questions for them, I wanted to know”.(Antonia)

#### 3.3.2. Care in the Present/without any Distractions and a Normalization of the Situation

The professionals developed tools to allow them to adapt to the new situation and to acquire the resilience necessary to deal with unforeseen and conflictive situations.

“Inside the hospital you had to remain focused every step of the way. This is what comes through the door, this is what I’ve got to do, and so this is what I’m doing. Trying to forget about everything else, that helped me a lot. Step by step, this is what I’ve got between my hands, and so this is what I have to figure out. And then, once I got home, I shut the world out, no news, no television”.(Jennifer)

With the passing of time, professionals experienced a certain normalization of the situation. One might say that they got used to the changes and adapted to them. It is likely that this is because of the necessary mechanism of emotional survival.

“I think so, the first few days, I was a bit nervous before going to work, but then you get used to it little by little, and then your days off help you to relax and relieve the fatigue and tension”.(Elisa)

#### 3.3.3. Demand for a Restoration of Best Practices

The midwives believe that situations that are very damaging for women have occurred as a consequence of the changes to the protocols. The protocol changes did not take a holistic view of the problem.

“... I think that with all this situation, with the pandemic, we’ve gone a bit backwards in terms of our healthcare models and treatment practices”.(Nancy)

Due to this situation, they have requested a return to face-to-face visits. Midwives and mothers see face-to-face treatment as something fundamental for the bonding process of the professional and the emotional wellbeing of the mother.

“I think that in-person visits will be back”.(Sara)

#### 3.3.4. Empowerment and Opportunities

Professionals have become aware of the power that they have, and of the good that they do and can do for their patients. We were able to observe how the professionals took pride in the good work that they were performing.

“... and then the renewed strength, you felt very strong because you’d been there, helping people who needed you, and well, when everything was “over” I felt really empowered”.(Alice)

Additionally, they have also become aware of the power that women have acquired thanks to the work that has been carried out in relation to reproductive rights.

“A group of women got together to vindicate their rights, lodge a complaint to show their disagreement with the fact that birth partners were not allowed into the delivery room”.(Antonia)

They have also detected opportunities, something that is quite normal in all crises, both in terms of exercising their profession and in terms of managing the health system.

Owing to the risks associated with transferring all COVID cases to hospitals, we should consider the benefits offered by maternity homes and homebirths:

“…... I believe that hospitals should be as separated as possible from maternity services”.(Aurora)

Other associated opportunities include reducing the time of admission and including home births in the publicly financed system, as these are not currently covered by the Spanish healthcare system.

“In my opinion, it would be ideal if we could create some low-risk units, run by midwives, or promote home births within the public health system for low-risk pregnancies; we must try to reduce the patient’s stay at the hospital as much as we can. We must encourage them to come only when they’re actually in labor and then check them out of the hospital as quickly as possible”.(Elisa)

## 4. Discussion

On the basis of the results obtained, a closer examination of the experiences of midwives on the frontline during the first months of the pandemic involves knowing and understanding a series of situations, experiences, and in particular, the conditions related to emotional, mental and social health. The midwives said that these aspects influenced their professional and personal performance in one way or another.

The first category identified through an analysis of the interviews refers to the cascade of emotions experienced by midwives. The emotions that midwives felt reflect a variety of the feelings that they experienced during the pandemic, both inside delivery rooms and on home care visits. This was particularly true in the case of those midwives who were transferred to other services to look after patients with COVID-19.

The emotions described by the interviewed participants ranged from fear to uncertainty, and also included worry and anxiety; we find a similar scenario in the meta-analysis of Shorey and Chan [28], which describes the negative psychological responses experienced by midwives and nurses during the pandemic, although it was observed that the family occupied a fundamental role in terms of providing support to the professionals. In our case, the family has occasionally been a source of worry for the midwives who witnessed members of their family taking ill or who saw how their partner had to stay in a hotel away from the home for fear of infecting the interviewed midwife who was pregnant. Aksoy and Koçak [29] identify a negative impact on work, the family and private life for professionals in addition to the psychological effects they experience. In this way, one can clearly see the support and efforts that are required at a global level to meet the challenges that midwives face when trying to deal with the emotional toll that the pandemic has left. In this regard, authors such as Bar-Zeev et al. [30] highlight the importance of having institutional backing or support from international bodies such as, for example, the UNFPA (United Nations Population Fund), because it is known that midwives play an essential role in the healthcare chain and that, as a result of the contingency, they find themselves working under a lot of tension, not just physical, but psychological as well. Thus, and in relation to what the interviewed midwives point out, the authors Bar-Zeev et al. [30] add that midwives often have to look after their families and their communities and put in a double working day, or what Estevan-Reina et al. [31] term the “double presence” factor, whereby total responsibility for care and wellbeing lies with women because of the need to respond and to devote themselves to the work, family and their areas of care, all at the same time. This increased the level of tension, stress and anxiety felt by some of the midwives that we interviewed. In this regard, when analyzing the interviews, we saw how the midwives mention confusion, work pressure, anxiety etc., in addition to fear, especially the fear of becoming infected. Likewise, some of the interviewed midwives refer to the insecurity that they felt because of a lack of knowledge about the disease and because of protocol changes, and this coincides with what other authors emphasize [32] when they say that professionals do not feel safe when healthcare guidelines are continually changing or are ambiguous. Protocols and guidelines issued within the work environment are closely linked to the occupational wellbeing of professionals.

Another of the relevant dimensions mentioned by the interviewed midwives is related to the professional occupation and concern for the women. In this regard, one of the complex experiences mentioned by the participants is linked to changes in the work functions and activities that they have traditionally carried out and which they have had to change as a consequence of COVID-19. In some cases, this situation did not involve any major transformations, although in others it was radically complex and even affected their occupational performance, as in the case of the midwives who had to change service and work as nurses. According to the accounts, this latter situation reflected the precariousness of the services in which they were working on the one hand, and on the other, interference in their bonds and their dealings with their patients and in providing them with appropriate care arising from the lack of knowledge or the fact that their experience was not up to date. Baumann et al. [33] touch on this aspect when referring to how independent midwives adapted because of the pandemic, sidestepping various questions and mainly creating anxiety about the future and concerns about how to guarantee the continuity and safety of care, in spite of the pandemic. In the same vein, the qualitative study of Ferreira et al. [34] points to two important variables that should be taken into account in these situations. One is related to the work overload that the professionals had to deal with, and on this point the women interviewed say that they were subjected to pressure at work because of changes to their timetables and services, although they did highlight the importance that permanent dialogue and companionship among their own colleagues has had in overcoming these complications. Another variable to be considered according to Ferreira et al. (2020) [34] is related to the daily risk of infection that health professionals are exposed to. In this regard, the fear of catching the virus was mentioned by the interviewees in almost all the cases and most especially in the case of those who saw a change in their everyday functions. Occasionally, this involved leaving the delivery wards and going to work elsewhere as COVID-19 nurses. On this matter, a study of the impact of COVID-19 on the practice of midwives in Kenya, Uganda and Tanzania shows how the pandemic has worsened the situation in countries with a shortage of health professionals, creating major challenges for professional activity. It also highlights the fact that midwives are faced with the paradox of having to look after other women while exposing themselves and their families to greater risks. There are even cases of infections between midwives and other health personnel [35], and this aspect is abundantly clear in our study. Furthermore, the ethical stress suffered during the pandemic is clear from the statements made by the participants. The midwives describe behaviors that would have been considered none too ethical in situations prior to the pandemic but which might be justified in such a situation.

Finally, the midwives interviewed recognize that, notwithstanding the complexity of the panorama—especially during the first months of the pandemic when there was not a lot of information available about healthcare procedure or about how the virus acts—their professional role became essential in this process, and this is outlined in the description of resistance and resilience strategies.

Health workers experienced a fear of infection during the pandemic, and they were afraid of infecting their family members, friends and colleagues. This resulted in feelings of uncertainty, stigmatization and anxiety as well as stress and emotional exhaustion, which resulted in dissatisfaction with their work. Nonetheless, one of the characteristics that the interviewed midwives highlighted about the pandemic was the opportunity that the social health crisis offered midwives in terms of assessing and redesigning the spaces for childbirth care, either in the child bearer’s home or in areas separated from the rest of the hospital. On this point, Davis-Floyd et al. [9] mention some concerns about changes to pregnancy and childbirth care as a result of the pandemic, especially considering the role of the midwives now that hospitals are being perceived as a source of contagion.

In turn, the interviewed midwives highlight the strength and empowerment that the pandemic has obliged them to generate as an element of resilience after assessing the power that they hold in their actions both in health management and administration and in caring for women, creating or improving strategies to provide proper care to the women they look after. An example of this can be found in the research undertaken by Liu et al. [36] describing the strategies devised by midwives and nurses to limit the transmission of COVID-19 among infected women in their third trimester of pregnancy, reduce the incidence of nosocomial infections, and promote safety in the care of the women and their babies. Another example is that provided in the study of Homer et al. [37] on independent midwives and the strategies that they have adapted with imagination, seeking information and adjusting their services to accompany home births. In our study, the midwives formed teams and drew up protocols, thus increasing companionship and compensating for the shortage of management teams brought about by the pandemic.

### Limitations

This study has some limitations related to the actual limitations of its own design. First of all, qualitative researchers must critically analyze the personal bias inherent in the design of studies of this type. To avoid this, the researchers of the study discussed the categories and the target categories that were defined. Secondly, the bias of the participants was considered. To avoid social desirability bias, semi-structured interviews were conducted with open questions asked in such a way that participants could feel free to answer without being judged by their reply. Thirdly, the sample was extracted, in part, through snowball sampling. In the selected sample, there is a representation of a large part of the various levels of the hospital and primary care centers of two Spanish autonomous communities, as well as of midwives providing homecare. Even though the results cannot be generalized to a wider segment of the population, the in-depth analysis carried out on the experiences of the midwives who were interviewed and the exhaustive description of both the sociodemographic and the contextual characteristics of the study make it possible to replicate and assess their transferability. Fourthly, we would like to emphasize the need to keep on studying the topic of research; notwithstanding the existence of incipient literature [38,39] in this regard, the interviews did not look at legal aspects linked to the professional liability of midwives.

## 5. Conclusions

Midwives experienced strong emotions during the early months of the COVID-19 pandemic. A fear of contagion was very present among themselves and their women patients.

In spite of the difficulties and the cascade of emotions they experienced, the midwives were concerned about the loss of women’s rights, the loss of autonomy and the increase in the level of vulnerability. There were difficulties at a professional level but there were also complications at a social and economic level, and it is difficult for working women to find a good work–life balance. This is especially true for midwives working on the frontline. The need for society and the political authorities to be aware of this situation is seen as something essential.

Resistance and resilience strategies were expressed in terms of the empowerment that the pandemic obliged them to generate. Midwives have become aware of the power that they have in their actions both in health management and administration and in providing care for women and in creating or improving the strategies that allow them to provide dignified care to their users.

Safety was a recurring theme in the management of the pandemic, and this led to a rethinking of childbirth care. Creating safe spaces where women can give birth with autonomy and dignity obliges us to continue thinking about and researching the spaces for childbirth care.

## Figures and Tables

**Figure 1 ijerph-18-06516-f001:**
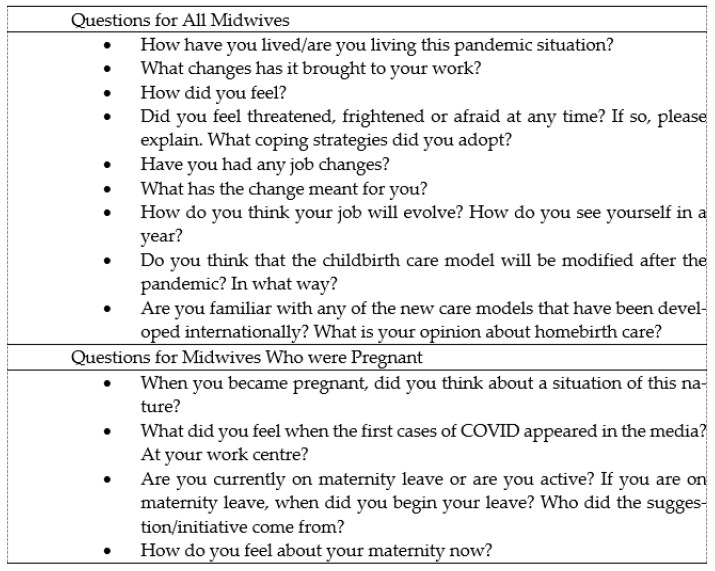
Interview outline.

**Figure 2 ijerph-18-06516-f002:**
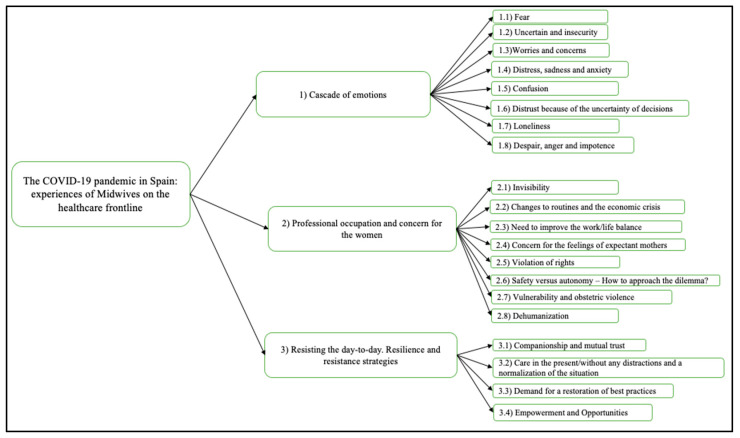
Categories based on our analysis of the transcriptions.

**Table 1 ijerph-18-06516-t001:** Profile of participating midwives.

Name	Age	Area of Work	COVID-19	Pregnant
Bárbara	30	Primary Care	No	Yes
Aurora	32	Hospital Delivery Room	No	Yes
Nancy	50	Hospital. Delivery Room Coordinator	Yes. Seriously ill. Admitted Hospital	No
Alice	37	Transferred to COVID ward	No	No
Antonia	42	Hospital. Delivery Room and Home-birth care	No	No
Elisa	45	Hospital. Delivery Room	No	No
Jennifer	44	Hospital. Delivery Room	No	No
Jane	45	Hospital. Delivery Room Coordinator	No	No
Maria	29	Providing home-birth care	Yes. Mildly ill. Did not have to be admitted to hospital	No
Sara	35	Primary Care	No	No

Table compiled by the authors. (The names of midwives have been changed to preserve the anonymity and confidentiality of the participants).

## Data Availability

The data presented in this study are available on request from the corresponding author. The data are not publicly available due to privacy restrictions.

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
