# Peer review of "The COVID-19 Pandemic in Spain: Experiences of Midwives on the Healthcare Frontline"

_ijerph, 2021, doi:10.3390/ijerph18126516_

Round 1
Reviewer 1 Report
The article presented is of great interest and relevance.
It is worth contemplating the lessons that can be learned from how the first wave of COVID was managed in Spain in order to implement good practices in the future.
I would also like to congratulate the authors for the methodological quality of the work carried out and for the presentation of the manuscript. And for choosing a qualitative methodology.
Below are some aspects that could be considered for improving the work presented:
- It would be worth explaining why the authors include in the table whether the midwives were pregnant or not and how that fact may influence their perception and experience of their work.
- It would be interesting if the authors could expand on the data analysis. It is one of the most relevant aspects of qualitative research and, in this case, it is very brief.
Author Response
Dear reviewer:
Thank you very much for your comments.
We send you a warm greeting and we respectfully attach our letter of rebuttal, which includes your suggestions as a reviewer and the modifications implemented by authors.
We have tried to respond to your requests. We hope they are satisfactory for you
Kind regards,
Josefina Goberna-Tricas, Ainoa Biurrun-Garrido, Carme Perelló-Iñíguez and Pía Rodríguez-Garrido.

Reviewer 2 Report
I read with interest the article entitled " The COVID-19 pandemic in Spain: Experiences of midwives on the healthcare frontline”
The study has the peculiarity of taking into consideration a group of midwives who worked in the frontline during the pandemic.
In order for the work to be considered for publication, some clarifications are needed.
Major Concerns
- The chapter dedicated to the results is not only not presented in an analytical way but reports the statements of the individual midwives. The results should be presented in a synthetic and analytical way. Reporting the name of the midwife together with the hospital where the study was done makes the subjects recognizable.
The whole chapter from lines 232 to 716 should be totally rewritten and the results should be presented analytically.
- Authors should specify the period in which the interviews were carried out and contextualize it with respect to the phase of the pandemic in their country at that time. [see Nioi, Matteo, et al. "COVID-19 and Italian healthcare workers from the initial sacrifice to the mRNA vaccine: Pandemic chrono-history, epidemiological data, ethical dilemmas, and future challenges." Frontiers in Public Health 8 (2020).]
- Some clarifications are needed on the methods.
- Has the questionnaire been validated?
- Has the estimate of the sample size been carried out? If not, why not?
- What was the legal context regarding the responsibility of healthcare professionals in Spain when the events described took place? [d'Aloja, Ernesto, et al. "COVID-19 and medical liability: Italy denies the shield to its heroes." EClinicalMedicine 25 (2020).; Nioi M, et al. Fear of the COVID-19 and medical liability. Insightsfrom a series of 130 consecutives medico-legal claims evaluated in a single institution during SARS-CoV-2-related pandemic.Signa Vitae. 2021. doi:10.22514/sv.2021.098]
Minor concerns
- Are there similar jobs for other categories of workers? The number of references should be implemented.
There are currently numerous articles on HCW's and COVID-19. The merit of the work is to describe the midwives' point of view but the chapter on "results" is written in a journalistic and unscientific way. The style of a paper should be different from that of an interview.
In order for the paper to be considered publishable it needs profound changes. I believe that the five points indicated above are essential, the most important of which is the presentation of the results.
Author Response
Dear Reviewer
We send you a warm greeting and we respectfully attach our letter of rebuttal, which includes your suggestions as a reviewer and our explanations and modifications implemented.
Kind regards,
Josefina Goberna-Tricas, Ainoa Biurrun-Garrido, Carme Perelló-Iníguez and Pía Rodríguez-Garrido

Round 2
Reviewer 2 Report
The authors met the requests of the first round. I think the paper has improved considerably and I have no further comments or requests to make.